# Floristic Diversity and Phytogeography of JABAL Fayfa: A Subtropical Dry Zone, South-West Saudi Arabia

**Ahmed M. Abbas** [1,2,*,†] , **Mohammed A. Al-Kahtani** [1], **Mohammad Y. Alfaifi** [1] ,
**Serag Eldin I. Elbehairi** [1,3] and **Mohamed O. Badry** [2,†]

1   Department of Biology, College of Science, King Khalid University, Abha 61413, Saudi Arabia;
    dr.malkahtani@gmail.com (M.A.A.-K.); alfaifi@kku.edu.sa (M.Y.A.); Serag@kku.edu.sa (S.E.I.E.)
2   Department of Botany & Microbiology, Faculty of Science, South Valley University, Qena 83523, Egypt;
    mohamedowis@svu.edu.eg
3   Cell Culture Lab., Egyptian Organization for Biological Products and Vaccines (VACSERA Holding
    Company), 51 Wezaret El-Zeraa St., Agouza, Giza 12311, Egypt
*   Correspondence: ahhassan@kku.edu.sa; Tel.: +966-540271385
†   These authors contributed equally as co-first authors.

**Abstract:** The present study surveyed the flora of the Jebel Fayfa region, South-West Saudi Arabia to analyze four elements of the vegetation: floristic diversity, life form, lifespan, and phytogeographical affinities. A total of 341 species of vascular plants were recorded belonging to 240 genera in 70 families, of which 101 species distributed among 40 families were considered as new additions to the flora of Jabal Fayfa. Six species are considered endemic to the study area while 27 are endangered. The most represented families were Fabaceae, Asteraceae, and Poaceae. The flora of Jabal Fayfa exhibited a high degree of monotypism. A total of 20 families (28.57%) were represented by a single species, and 180 genera (75.00%) were monotypic. The recorded flora consists of 70.09% perennials and 29.91% annuals. Phanerophytes and therophytes were the most frequent lifeforms. Phytogeographical analysis revealed that the biregional elements of the Saharo-Arabian/Sudano-Zambezian chorotype are the most dominant chorotypes (35.48%), forming two-thirds of the floristic structure in Jabal Fayfa. The new additions to the local flora of the region indicate that the Jabal Fayfa region and the country need further thorough botanical exploration and documentation which would help in adding several species to the flora of Saudi Arabia.

**Keywords:** angiosperms; endemism; Fayfa Mountain; flora; life form; new records

## 1. Introduction

The Kingdom of Saudi Arabia covers about two-thirds of the Arabian Peninsula, with an area of about two million square kilometers in extent, comprising a variety of distinct physiographical habitats, such as mountains, valleys ('wadis'), sandy and rocky deserts, meadows ('raudhas'), lava areas ('harrats'), and salt pans ('sabkhahs'), with a natural wealth of plant species [1–3].

The flora of the Kingdom contributes to one of the richest biodiversity hot spots in the Arabian Peninsula, comprising important genetic resources of medicinal plants, xerophytic vegetation, and crops [4]. It comprises about 2290 species and 855 genera (including pteridophytes and gymnosperms) in 131 families, with a number of endemic species, about 200 regional endemics (2.5% of the total flora) [2,5].

The components of the flora of Saudi Arabia is somewhat a complex, having affinities with the floras of North Africa, East Africa, the Mediterranean, and Irano-Turanian countries [1,6–8].

The north-western and south-western territories of Saudi Arabia harbor a rich flora and contain the highest number of species, about 80% of the total flora of the country [2]. The south-western region is of great interest from the floristic and phytogeographic point of view because it represents a link between Asia and Africa continents [9–14]. Moreover, the flora of the extreme southwest mountains has the greatest plant species diversity in the Kingdom of Saudi Arabia, due to a large annual rainfall and the range of altitude from sea level to 3100 m [14].

Jabal Fayfa (also known as Fayfa or the Faifa Mountains) in Jizan Province is an important plant diversity hotspot of southwestern Saudi Arabia [2]. It is characterized by a mosaic of environments and a variety of habitats, and harbors a rich and diverse flora [15]. Several studies on the floristic composition and vegetation diversity of different localities, including mountains, wadies, plains, and islands in the Jizan region, southwestern Saudi Arabia have been undertaken [5,15–34]. However, only a few studies focused on the vascular flora and plant ecology of Jabal Fayfa [35,36]. This is perhaps due to its wide area, the range of climates, rocky topography, and lack of vehicular access roads along the mountainous escarpment of this region which has resulted in a paucity of floristic studies and no complete survey of the flora of this region.

To best of our knowledge, there are no earlier reports on the flora of Jabal Fayfa which reflect the exciting range of environmental conditions, leading us to recognize that more floristic work was needed to fill gaps in our understanding of this flora. The current study aims to survey and identify the floristic diversity, lifespan, life forms, phytogeographic relationships, and update the checklist of the wild plants growing in the Jabal Fayfa region.

## 2. Materials and Methods

### 2.1. Study Area

This study was conducted in Jabal Fayfa area (17°14′ N–43°05′ E) in the southwestern region of Jizan, Saudi Arabia (Table 1, Figure 1). These mountains are not a single steep ridge, but a series of mountain stretches varying in elevation from 900 m to about 2000 m cut by deep valleys and extending over several kilometers along a roughly north-south axis. On the northern side, they join the mountains of Bani Malik, and on the South, they link with the mountains of Yemen [35].

**Table 1.** Locations of the studied areas in Jabal Fayfa showing the sampling sites with their coordinates, number of stands, and elevation.

| Locations | No. of Stands | Coordinates | | Elevation (m) |
|:---:|:---:|:---:|:---:|:---:|
| | | Latitude (N) | Longitude (E) | |
| 1 | 10 | 17°25′14.0″ N | 43°08′02.0″ E | 1700 |
| 2 | 8 | 17°21′50.0″ N | 43°17′29.0″ E | 1850 |
| 3 | 6 | 17°24′43.0″ N | 43°06′29.0″ E | 900 |
| 4 | 5 | 17°14′48.0″ N | 43°09′45.0″ E | 850 |
| 5 | 9 | 17°13′34.0″ N | 43°10′12.0″ E | 321 |
| 6 | 7 | 17°15′06.0″ N | 43°11′15.0″ E | 800 |
| 7 | 6 | 17°16′29.0″ N | 43°11′35.0″ E | 700 |
| 8 | 8 | 17°17′15.0″ N | 43°13′31.0″ E | 700 |
| 9 | 6 | 17°17′13.0″ N | 43°14′31.0″ E | 830 |
| 10 | 5 | 17°20′02.0″ N | 43°18′40.0″ E | 1970 |
| 11 | 7 | 17°21′35.0″ N | 43°20′37.0″ E | 2200 |
| 12 | 5 | 17°27′21.0″ N | 43°06′00.0″ E | 1780 |
| 13 | 9 | 17°23′38.4″ N | 43°12′52.6″ E | 1400 |
| 14 | 6 | 17°19′59.0″ N | 43°15′11.8″ E | 900 |
| 15 | 8 | 17°15′34.9″ N | 43°16′44.6″ E | 940 |
| 16 | 7 | 17°25′29.0″ N | 43°11′00.2″ E | 1400 |
| Total | 112 | | | |

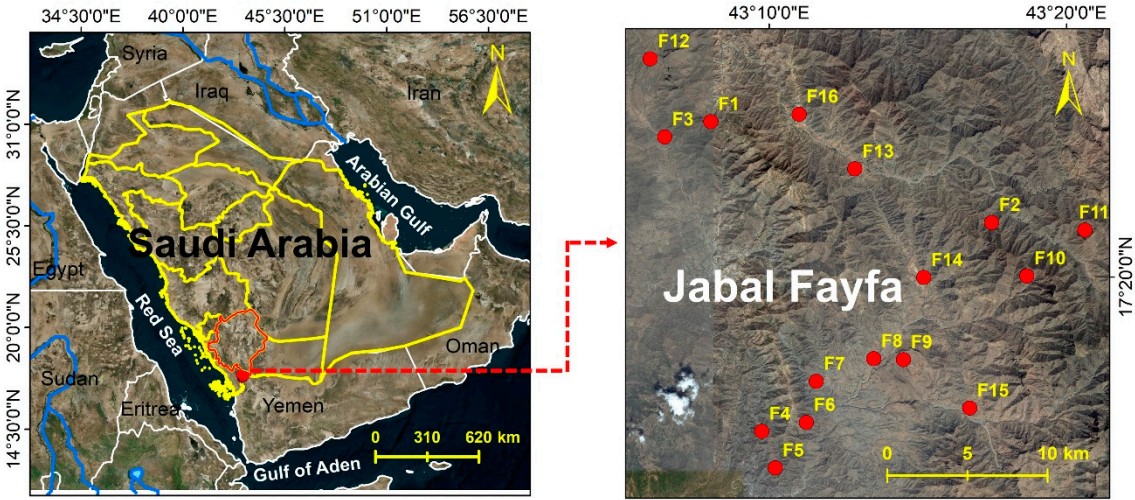

**Figure 1.** Location map of the study area of Jabal Fayfa showing the sampling sites (red).

The study area lies within the subtropical dry zone which is characterized by hot summers and warm winters [37]. The total annual rainfall during the last 20 years in the field (January 2000–August 2020) was 1560 mm, being concentrated mainly in July and August. The average monthly maximum air temperature was 34.9 ± 0.93 °C, varying between 30.6 °C in November and 38.3 °C in June and July, while the average monthly minimum air temperature was 26.9 ± 0.88 °C, varying between 22.9 °C in January and 31.0 °C in July (Jizan City Meteorological Station, 7 m above sea level; 16°53′48.5″ N 42°35′02.4″ E) (Figure 2).

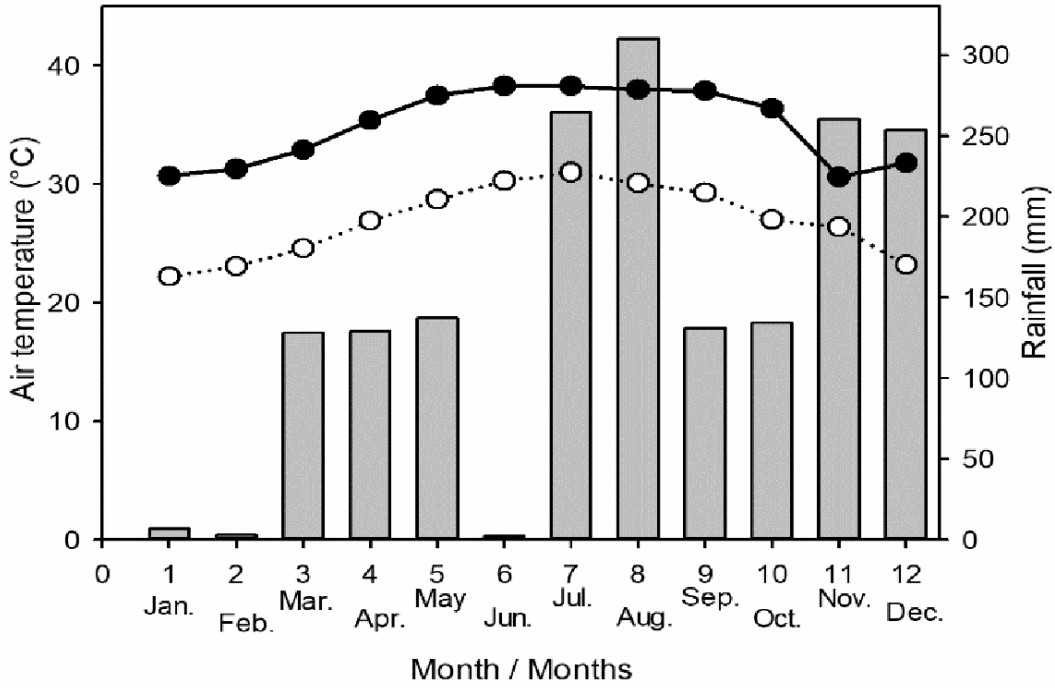

**Figure 2.** Gaussen diagram showing monthly rainfall (mm; bars) and maximum (black circles) and minimum (open circles) monthly air temperature (°C) at Jizan city nearby Jabal Fayfa (Southeast Saudi Arabia) during the last 20 years in the field (January 2000–August 2020).

## 2.2. Plant Collection and Species Identification

Field collections were made by the first author (A. M. Abbas) at different times during intensive floristic surveys of wild populations in the study area between January–December 2019. A total of

16 localities were selected in the study area, which were divided into 112 stands (vegetation plots). In each location, sampling stands were situated randomly using the Reléve method [38]. In selecting each locality and stand, a reasonable degree of plant cover homogeneity, physiographic variation and habitat uniformity were ensured (Figure 1).

The collected plant species were identified and named according to the available literature [39–46], and were updated according to [46]. Plant life-forms along with life span were determined [47–49] and phytogeographical affinities of the surveyed species were defined [10,50–53]. Specimens were dried and deposited in the herbarium of the Biology Department, College of Science, King Khalid University, Saudi Arabia.

## 3. Results

### 3.1. Floristic Composition

A total of 341 taxa of vascular plants were recorded from the study area, belonging to 240 genera in 70 families. Among them, 101 species (about 29.62% of the total flora surveyed) have been recorded for the first time and represent new additions to the flora of Jabal Fayfa, based on the earlier flora records from this region [2,35,36]. These new records are distributed among 40 families. Moreover, six species were considered endemic to the study area, of them five are endemic-not endangered (*Anisotes trisulcus* (Forssk.) Nees, *Barleria bispinosa* (Forssk.) Vahl, *Barleria bispinosa* (Forssk.) Vahl, *Ceropegia aristolochioides* Decne, and *Reseda sphenocleoides* Deflers), and one is endemic-endangered (*Aloe woodii* Lavranos and Collen). On the other hand, 27 species considered endangered of extinction and 18 species were invasive alien taxa (Appendix A).

Angiosperms were represented by 69 families, of which 63 families (90%) were dicotyledons with 293 taxa (85.92%), while monocotyledons were represented by 6 families (8.57%) and 47 taxa (13.78%). One family belongs to the gymnosperms, Cupressaceae, and was represented by only one species (*Juniperus procera* Hochst. ex Endl.) representing 0.29% of the survey.

Fabaceae (38 species = 11.14%), Asteraceae (31 species = 9.09%), and Poaceae (30 species = 8.08%), were the most species-rich families. Amaranthaceae was represented by 17 species (4.99%). Acanthaceae, Apocynaceae, Euphorbiaceae, and Lamiaceae were represented by 14 species each (4.11%), while Malvaceae and Boraginaceae were represented by 13 species (3.81%) and 10 species (2.93%), respectively. Cucurbitaceae, Moraceae, and Solanaceae were represented by 7 species each (2.05%), while Commelinaceae, Cyperaceae, Nyctaginaceae, and Scrophulariaceae were represented by 5 species each (1.47%). Cleomaceae, Crassulaceae, Verbenaceae, and Vitaceae were represented by 4 species each (1.17%). Eleven families were represented by three species (0.88%), meanwhile, 18 families were represented by two species (0.59%). On the other hand, 20 families were poorly represented, having one species each (0.29%). The largest families in terms of the number of genera were Asteraceae (25 genera), Fabaceae (23 genera), and Poaceae (21 genera) (Appendix A, Figure 3).

The genera with the larger number of species were *Euphorbia* L. with eight species (2.35%), *Ficus* Tourn. ex L. with seven species (2.05%), *Indigofera* L. with six species (1.76%), *Amaranthus* L., *Cenchrus* L., *Cleome* L., *Cyperus* L., *Solanum* L., and *Vachellia* Wight and Arn. with four species each (1.17%). According to the duration or life span, most of the species recorded during this survey were perennials with 239 species (70.09%) of the total recorded species, while the annuals were represented by 102 species (29.91%) (Appendix A).

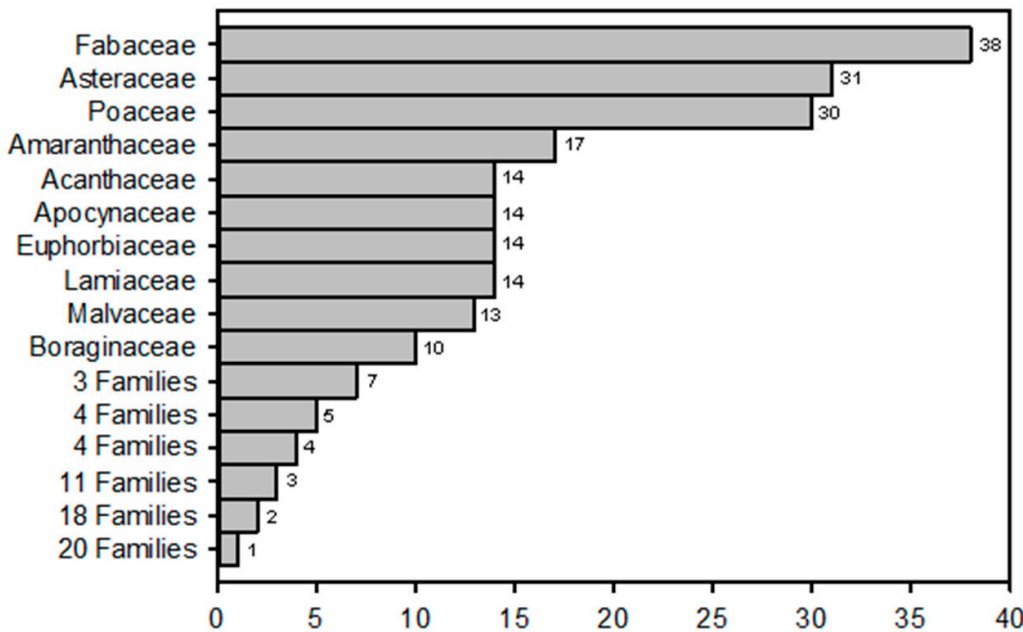

**Figure 3.** Histogram of the floristic composition of the 70 families surveyed in Jabal Fayfa.

## 3.2. Life-Form Spectra

Seven life forms were recorded in the current study. The plant life form classes along Jabal Fayfa indicated that Phanerophytes were the most frequent life form (103 species = 29.91%), followed by Therophytes (100 species = 30.21%), Chamaephytes (86 species = 25.22%), Hemicryptophytes (45 species = 13.20%), Parasites (4 species = 1.17%), and Geophytes with (2 species = 0.59%). While Geophytes-Helophytes were represented by a single species *Cyperus alternifolius* subsp. *flabelliformis* Kük. (0.29%) (Appendix A, Figure 4).

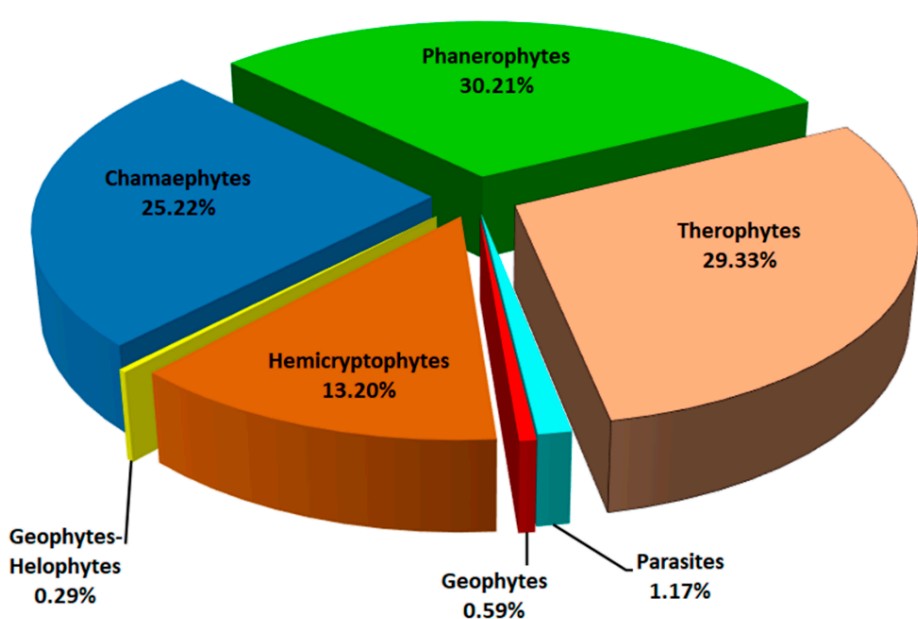

**Figure 4.** Life form spectrum of plant species recorded in Jabal Fayfa.

## 3.3. Chorological Affinities

Chorological analysis of the 341 plant species recorded in this study classified them into three major phytogeographical groups: monoregional, biregional, and pluriregional. A total of 42 species

representing 12.32% of the total number of recorded species were monoregional taxa of different affinities. The recorded monoregional elements fall under four main chorotypes: Saharo-Arabian taxa (21 species forming 6.16% of recorded species), and Sudano-Zambezian taxa (18 species forming 5.28% of recorded species). Two Australian taxa were recorded in the study area (*Asystasia gangetica* (L.) Anderson and *Dysphania carinata* (R.Br.) Mosyakin and Clemants) representing 0.59% of the surveyed flora. The last chorotype (Deccan) was rarely represented in the study area with only one species (*Dichrostachys cinereal* (L.) Wight and Arn.) forming 0.29% of the total number of plant species surveyed.

The biregional elements were the highest represented (139 species = 40.76%) among the surveyed species in the study area with different affinities. The recorded biregional elements fall under four main chorotypes: the Saharo-Arabian/Sudano-Zambezian chorotypes together have the highest share of species (121 species), representing 35.48% of the total flora surveyed, followed by the Saharo-Sindian/Sudano-Zambezian region, represented by 14 species (4.11%). While the lowest share of species was recorded for the Mediterranean/Saharo-Arabian and the Irano-Turanian/Saharo-Arabian regions with two species (*Aloe vera* (L.) Burm.f. and *Hypochoeris glabra* L., 0.59%) and one species (*Phoenix dactylifera* L., 0.29%), respectively.

The pluriregional elements were represented by a total of 68 species (19.94%) of different affinities. These pluriregional species fall under eight main chorotypes: Irano-Turanian/Mediterranean/Saharo-Arabian/Sudano-Zambezian (22 species forming 6.45% of recorded species), Irano-Turanian/Saharo-Arabian/Sudano-Zambezian (18 species forming 5.28% of recorded species), and Irano-Turanian/Mediterranean/Saharo-Sindian/Sudano-Zambezian (10 species forming 2.93% of recorded species). Both Mediterranean/Saharo-Arabian/Sudano-Zambezian and Irano-Turanian/Saharo-Sindian/Sudano-Zambezian regions were represented by seven species (2.05% of recorded species). Irano-Turanian/Mediterranean/Saharo-Sindian region was represented by two species (*Capparis spinosa* var. *aegyptia* (Lam.) Boiss. and *Malva parviflora* L., 0.59%), while only one species (0.29%) occurred in the following regions: Mediterranean/Saharo-Sindian/Sudano-Zambezian (*Sisymbrium erysimoides* Desf.) and Irano-Turanian/Mediterranean/Saharo-Arabian (*Onopordum heteracanthum* C.A. Mey.). The remaining 92 species were distributed among Palaeotropical (35 species = 10.26%), Neotropical (26 species = 7.62%), Pantropical (20 species = 5.87%), and cosmopolitan (12 species = 3.52%) chorotypes (Table 2, Figure 5).

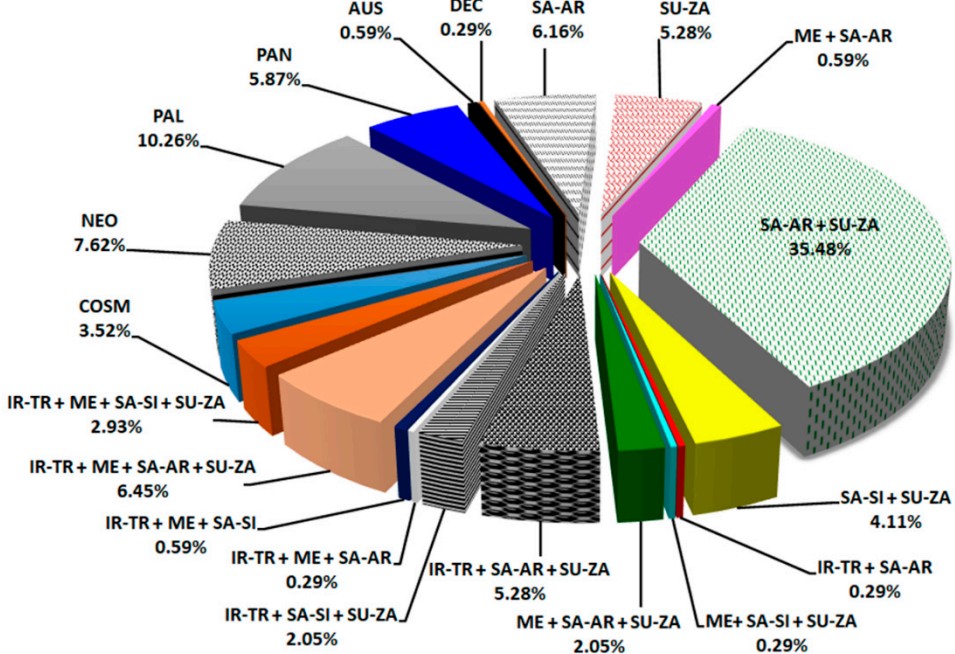

**Figure 5.** Phytogeographical analysis of the recorded species in Jabal Fayfa. For the abbreviations see Appendix A.

**Table 2.** Numbers of plant species belonging to the main floristic chorotypes and their relevant percent (%) recorded in Jabal Fayfa.

| Chorotype | No. of Plant Species | Percentage (%) |
|---|---|---|
| Cosmopolitan | 12 | 3.52 |
| Neotropical | 26 | 7.62 |
| Palaeotropical | 35 | 10.26 |
| Pantropical | 20 | 5.87 |
| Total | 92 | 26.98 |
| Monoregional | | |
| AUS | 2 | 0.59 |
| DEC | 1 | 0.29 |
| SA-AR | 21 | 6.16 |
| SU-ZA | 18 | 5.28 |
| Total | 42 | 12.32 |
| Biregional | | |
| ME + SA-AR | 2 | 0.59 |
| SA-AR + SU-ZA | 121 | 35.48 |
| SA-SI + SU-ZA | 14 | 4.11 |
| IR-TR + SA-AR | 1 | 0.29 |
| Total | 139 | 40.76 |
| Pluriregional | | |
| ME + SA-SI + SU-ZA | 1 | 0.29 |
| ME + SA-AR + SU-ZA | 7 | 2.05 |
| IR-TR + SA-AR + SU-ZA | 18 | 5.28 |
| IR-TR + SA-SI + SU-ZA | 7 | 2.05 |
| IR-TR + ME + SA-AR | 1 | 0.29 |
| IR-TR + ME + SA-SI | 2 | 0.59 |
| IR-TR + ME + SA-AR + SU-ZA | 22 | 6.45 |
| IR-TR + ME + SA-SI + SU-ZA | 10 | 2.93 |
| Total | 68 | 19.94 |

Chorotypes abbreviations: AUS: Australian, COSM: Cosmopolitan, DEC: Deccan, ME: Mediterranean, NEO: Neotropical, PAL: Palaeotropical, PAN: Pantropical, IR-TR: Irano-Turanian, SA-AR: Saharo-Arabian, SA-SI: Saharo-Sindian, SU-ZA: Sudano-Zambezian.

## 4. Discussion

The floral diversity of Jabal Fayfa is rich, including numerous valuable plant species. In the current study, a floristic analysis of vascular plant species from the study area includes 341 species belonging to 240 genera and 70 families (Appendix A). Compared to the floristic composition of different hotspot regions in Jizan of Saudi Arabia, species numbers recorded in this study were very high [26,29–31,34,54]. However, the number of species recorded in this study (341 taxa) is within the range of the flora recorded previously by [35]. This is thought to be mainly due to the presence of a mosaic environment in these undulating mountain ranges cut by deep valleys, forming a distinct number of habitats each with particular features regarding soil composition, topographic differences, water resources and urbanization activities.

Despite the large number of species recorded in the study area, the share of endemism is too little. Only six species (1.76%) are endemic to the Study area. Moreover, 27 species are considered endangered of extinction. The flora of Jabal Fayfa mountains are like that of the neighboring countries such as Yemen. The presence of endemic and endangered species in the local flora of Jabal Fayfa might be caused by the constant moisture-laden breezes from the Red Sea on the west-facing slopes, which hold several micro hotspots suitable to sustain these species [2]. On the other hand, 18 invasive alien taxa were recorded from the study area (Appendix A). These taxa have distressing impacts on native biota, causing decline or even extinction of some indigenous populations. As Jabal Fayfa lies to the border with Yemen. The major invasions of the flora of the area are due to intentional or

unintentional actions of humans, animals, birds, and to some extent, due to traffic across the borders. Nevertheless, most introductions have only minor impacts on the ecosystem [2].

The floristic survey of Jabal Fayfa showed that Fabaceae (38 species), Asteraceae (31 species), and Poaceae (30 species), were the most species-rich families, constituting the main bulk of plant species (29.03%). These results conformed to those of [1] for the flora of Saudi Arabia, and [35] for the flora of the Jabal Fayfa region. Moreover, the former three large families were reported as the most dominant in the flora of the Mediterranean, North Africa, eastern Ethiopia, and northern Zambia [55–57]. This can be attributed to the dispersal of their efficient seeds' capabilities, migration efficiency, and wide ecological range of tolerance, in addition to local conditions of water availability and depth. As in most subtropical and tropical deserts, a significant feature of the floristic composition of the flora of Saudi Arabia is that a few families are floristically rich [58].

The present study revealed that the floristic composition of Jabal Fayfa exhibited a high degree of monotypism. Among 70 families recorded, 20 families (28.57%) were represented by a single species. Moreover, 180 genera (75.00%) were monotypic. These results are in line with the findings of [2] who reported the presence of 33 monotypic families (25.19%) of the total number of families recorded in the entire flora of Saudi Arabia. This may be due to the fact that a flora of an area, in which the species are distributed among various genera, families or other higher ranks, exhibit greater genomic information and phylogenetic diversity than that in which the most species belong to the same genus or concentrated into fewer higher-ranks [59,60].

Moreover, taxonomic diversity in Jabal Fayfa is 1.42 (341/240) species per genus, a ratio less than 2.68 (2290/855) which was recorded in the total area of Saudi Arabia [2]. This great diversity may be due to the mosaic environment, the climatic variation, substrate discontinuities, abundance of water resources, and the diversity of topography [15,61,62].

Unlike the flora of other regions of the Kingdom, two-thirds (70.09%) of the flora of the Jabal Fayfa is represented by perennials while annuals were represented by 29.91% of the total flora recorded. This trend matches the finding of [15,35]. The dominance of perennial species in the plant cover defines the character of the vegetation. This may be attributed to the rather low rainfall, which is not enough for the growth of many annuals. On the other hand, perennial plants are adapted to the extreme habitats of the area, which then offers a characteristic physiognomy to the vegetation [63–66].

The Jabal Fayfa flora exhibits a great diversity of life forms. The life-form spectrum is predominantly phanerophytes (30.21%), followed by therophytes (29.33%) and chamaephytes (25.22%). The dominance of phanerophytes, therophytes, and chamaephytes over other life forms may be a response to the hot dry climate, variation in rainfall, topography, and landform in addition to human and animal interference [65,67,68]. Therophytes are characterized by their short life cycle and high growth rate which enables them to resist substrate instability and biotic influence, their ability to release copious seeds, their ecological, genetic and morphological plasticity under a high level of disturbance, a hot dry climate, lack of rainfall, topographic variation, and biotic influences [69–74]. This trend of life form spectra is similar to that of other regions of Saudi Arabia [57,63,64,75].

Chorological analysis of the 341 species surveyed in the study area revealed that the biregional elements of the Saharo-Arabian/Sudano-Zambezian chorotype (35.48%) are the most dominant chorotypes, forming the major constituent (about two-thirds of the recorded plants) of the floristic structure in Jabal Fayfa. Similar results were obtained in different studies of the flora of Saudi Arabia [21, 26,29,31,62,74,75]. However, the dominance of the biregional Saharo-Arabian/Sudano-Zambezian chorotype (35.48%) over the pluri- and monoregional chorotypes (19.94% and 12.32%, respectively) disagrees with the findings of [76].

Wickens [77] recognized five Domains (subregions) within the Sudano-Zambezian region; of these the South Arabian Domain is an extension of the Sudano-Zambesian including parts of southern Saudi Arabia and Yemen bordering the coasts of the Red Sea and the Gulf of Aden. Moreover, [7] suggested the extension of the Sudano-Zambezian region into southern and western Saudi Arabia.

Our results indicate that the percentage of the Mediterranean and Irano-Turanian elements within the bi- and pluri-regional chorotypes decrease southward (Table 2). This may be because the Mediterranean and Irano-Turanian chorotypes indicate the more mesic environment, while the Saharo-Arabian/Sudano-Zambezian chorotypes are good indicators of a desert environment, and therefore decrease moving south to be replaced by the Saharo-Arabian/Sudano-Zambezian elements [75,78]. This combination of different phytochoria with uneven numbers of species can be attributed to different factors such as diversity of habitats, topography, water availability and the capability of certain taxa to penetrate the study area from different adjacent phytogeographical regions.

Interestingly, the botanical exploration in Jabal Fayfa in the southwestern region of Saudi Arabia revealed that, out of the 341 taxa recorded, 101 species were considered new additions to the local flora of the region (Appendix A). It is worth noting that all the 101 new species recorded here were already known from other regions in Saudi Arabia. So, the addition of these new distributional records increases the total number of local plants known from Jabal Fayfa region to 638 taxa, a 19% increase from the 537 taxa previously recorded [2].

The report of these new taxa to the local flora of Jabal Fayfa can be attributed to the following factors: (i) the scarcity of new and/or up-to-date botanical explorations of this region, and (ii) the lack of vehicular access roads along the mountain ranges, (iii) the existence of abandoned agricultural terraces within the study area which possess secondary vegetation as a resulted in the presence of seeds of ruderal weeds within the crop seeds, (iv) the rich and diverse flora of the study area owing to the combination of East African, Mediterranean and Irano-Turanian species [7,8], (v) the unique geographical location of Jabal Fayfa as the region borders one of the botanically rich regions like Yemen, and (vi) the mosaic environment of the undulating mountains ranges of Jabal Fayfa making many regions remain underexplored.

This project has not surveyed all the area of Jabal Fayfa, however, the report of these new additions to the local flora of the region indicates that the Jabal Fayfa region and the surrounding country needs further thorough botanical exploration and documentation which would help in adding species records to the flora of Saudi Arabia.

## 5. Conclusions

The current study found that the floristic composition of Jabal Fayfa is highly diverse in comparison with many other regions in southwest Saudi Arabia. This diversity may be due to the combination of various environmental factors which are favorable for a wide range of plant species. A total of 341 taxa was recorded in the study area with 101 species being new additions to the local flora of the region, increasing the number of plants known from the Jabal Fayfa region to 638 taxa. This represents a 19% increase on the previously published species list and indicates the need for further thorough botanical exploration.

**Author Contributions:** Conceptualization, A.M.A., M.A.A.-K., M.Y.A., S.E.I.E. and M.O.B.; methodology, A.M.A., M.A.A.-K., M.Y.A., S.E.I.E. and M.O.B.; formal analysis, A.M.A. and M.O.B; writing—original draft preparation, A.M.A. and M.O.B.; writing—review and editing A.M.A.; funding acquisition, A.M.A. All authors have read and agreed to the published version of the manuscript.

**Funding:** This research was funded by Deanship of Scientific Research at King Khalid University, through General Research Project under grant number G.R.P.91-41.

**Acknowledgments:** We would like to thank to M.F., Australian National Herbarium and Australian National Botanic Gardens, Canberra, Australia, for his diligent proofreading and comments which improved this manuscript, S.K.A., Soils and Water Use Department, Agricultural and Biological Research Division, National Research Centre (NRC), Dokki, Egypt, for preparing the map of the study area.

**Conflicts of Interest:** The authors declare no conflict of interest.

# Appendix A

List of plant species recorded in Jabal Fayfa along with their families, life span, life form, and chorotypes.

| Family | Taxa | Life Span | Life Form | Chorotype |
|---|---|---|---|---|
| Acanthaceae | [+] *Anisotes trisulcus* (Forssk.) Nees | Per. | Ch. | SA-AR + SU-ZA |
| | [*,#] *Asystasia gangetica* (L.) T.Anderson | Per. | Ph. | AUS |
| | [+] *Barleria bispinosa* (Forssk.) Vahl | Per. | Ph. | SA-AR |
| | [+] *Barleria trispinosa* (Forssk.) Vahl | Per. | Ch. | SA-AR |
| | [*] *Blepharis edulis* (Forssk.) Pers. | Per. | Ch. | IR-TR + SA-AR + SU-ZA |
| | [#] *Blepharis maderaspatensis* (L.) B.Heyne ex Roth | Per. | Ch. | SA-AR + SU-ZA |
| | *Crossandra johanninae* Fiori | Per. | Ch. | SA-AR + SU-ZA |
| | *Dicliptera paniculata* (Forssk.) I.Darbysh. | Ann. | Th. | PAL |
| | [*] *Ecbolium gymnostachyum* (Nees) Milne-Redh. | Per. | Ch. | SA-AR + SU-ZA |
| | *Ecbolium viride* (Forssk.) Alston | Per. | Ph. | PAL |
| | *Hypoestes forskaolii* (Vahl) R.Br. | Per. | He. | SA-AR + SU-ZA |
| | *Justicia flava* (Forssk.) Vahl | Per. | Ch. | PAL |
| | [*] *Justicia heterocarpa* T.Anderson | Ann. | Th. | SA-AR + SU-ZA |
| | [*,#] *Lepidagathis scariosa* Nees | Per. | He. | SA-AR + SU-ZA |
| Aizoaceae | [*] *Sesuvium verrucosum* Raf. | Per. | He. | NEO |
| | [*,‡] *Trianthema portulacastrum* L. | Ann. | Th. | PAN |
| | *Zaleya pentandra* (L.) C.Jeffrey | Ann. | Th. | IR-TR + SA-AR + SU-ZA |
| Aloaceae | *Aloe fleurentiniorum* Lavranos & L.E.Newton | Per. | Ch. | SA-AR |
| | [*] *Aloe vera* (L.) Burm.f. | Per. | He. | ME + SA-AR |
| | [^] *Aloe woodii* Lavranos & Collen. | Per. | He. | SA-AR |
| Amaranthaceae | *Achyranthes aspera* L. | Ann. | Th. | PAN |
| | *Aerva javanica* (Burm.f.) Juss. ex Schult. | Per. | Ch. | PAL |
| | *Aerva lanata* (L.) Juss. Ex Schult. | Per. | Ch. | PAL |
| | [‡] *Alternanthera pungens* Kunth | Ann. | Th. | NEO |
| | *Amaranthus graecizans* L. | Ann. | Th. | PAL |
| | [‡] *Amaranthus hybridus* L. | Ann. | Th. | PAN |
| | [‡] *Amaranthus spinosus* L. | Ann. | Th. | NEO |
| | *Amaranthus viridis* L. | Ann. | Th. | COSM |
| | [#] *Celosia trigyna* L. | Ann. | Th. | SA-AR + SU-ZA |
| | *Chenopodiastrum fasciculosum* (Aellen) Mosyakin | Ann. | Th. | SU-ZA |
| | *Chenopodium album* L. | Ann. | Th. | COSM |
| | [*] *Chenopodium pratericola* Rydb. | Ann. | Th. | NEO |
| | *Digera muricata* (L.) Mart. | Ann. | Th. | PAL |
| | *Dysphania ambrosioides* (L.) Mosyakin & Clemants | Ann. | Th. | COSM |
| | *Dysphania carinata* (R.Br.) Mosyakin & Clemants | Ann. | Th. | AUS |
| | *Dysphania schraderiana* (Schult.) Mosyakin & Clemants | Ann. | Th. | SA-AR + SU-ZA |
| | *Pupalia lappacea* (L.) Juss. | Per. | Ch. | PAL |
| Anacardiaceae | *Searsia retinorrhoea* (Steud. ex Oliv.) Moffett | Per. | Ph. | SA-AR + SU-ZA |
| Annonaceae | [#,‡] *Annona squamosa* L. | Per. | Ph. | NEO |
| Apiaceae | [*] *Pimpinella menachensis* Schweinf. ex H.Wolff | Ann. | Th. | SU-ZA |

| Family | Taxa | Life Span | Life Form | Chorotype |
|---|---|---|---|---|
| **Apocynaceae** | *Adenium obesum* (Forssk.) Roem. & Schult. | Per. | Ph. | SA-AR |
| | *Calotropis procera* (Aiton) W.T.Aiton | Per. | Ch. | SA-AR + SU-ZA |
| | * *Caralluma subulata* (Forssk.) Decne. | Per. | Ch. | SA-AR |
| | *Carissa spinarum* L. | Per. | Ph. | PAN |
| | * *Catharanthus roseus* (L.) G.Don | Ann. | Ch. | SU-ZA |
| | [+] *Ceropegia aristolochioides* Decne. | Per. | Ge. | SA-AR + SU-ZA |
| | * *Cynanchum forskaolianum* Meve & Liede | Per. | Ph. | SA-AR |
| | * *Desmidorchis penicillata* (Deflers) Plowes | Per. | Ch. | SA-AR + SU-ZA |
| | *Desmidorchis retrospiciens* Ehrenb. | Per. | Ch. | SA-AR + SU-ZA |
| | * *Gomphocarpus fruticosus* (L.) W.T.Aiton | Per. | Ch. | SA-AR + SU-ZA |
| | *Kanahia laniflora* (Forssk.) R.Br. | Per. | Ch. | SA-AR + SU-ZA |
| | *Leptadenia arborea* (Forssk.) Schweinf. | Per. | Ph. | SA-AR + SU-ZA |
| | * *Monolluma quadrangula* (Forssk.) Plowes | Per. | Ch. | SA-AR |
| | * *Pentatropis nivalis* (J.F.Gmel.) D.V.Field & J.R.I.Wood | Per. | Ph. | IR-TR + SA-AR + SU-ZA |
| **Arecaceae** | * *Phoenix caespitosa* Chiov. | Per. | Ph. | SA-AR + SU-ZA |
| | * *Phoenix dactylifera* L. | Per. | Ph. | IR-TR + SA-AR |
| **Asparagaceae** | *Asparagus africanus* Lam. | Per. | Ph. | SA-SI + SU-ZA |
| | *Dipcadi viride* (L.) Moench | Per. | Ge. | SA-AR + SU-ZA |
| **Asteraceae** | *,[#] *Ageratum conyzoides* L. | Ann. | Th. | NEO |
| | *Baccharoides schimperi* (DC.) Isawumi, El-Ghazaly & B.Nord. | Per. | Ph. | SA-AR + SU-ZA |
| | * *Bidens bipinnata* L. | Ann. | Th. | NEO |
| | *Bidens pilosa* L. | Ann. | Th. | NEO |
| | *Conyza stricta* Willd | Per. | Ph. | IR-TR + SA-AR + SU-ZA |
| | *Crepis rueppellii* Sch.Bip. | Ann. | Th. | SA-AR + SU-ZA |
| | *Cyanthillium cinereum* (L.) H.Rob. | Ann. | Th. | PAL |
| | [‡] *Eclipta prostrata* (L.) L. | Ann. | Th. | NEO |
| | *Erigeron bonariensis* L. | Ann. | Th. | NEO |
| | * *Helichrysum foetidum* Moench | Ann. | Th. | SA-AR + SU-ZA |
| | *Helichrysum glumaceum* DC. | Ann. | Th. | SA-AR + SU-ZA |
| | * *Hypochoeris glabra* L. | Ann. | Th. | ME + SA-AR |
| | * *Kleinia odora* (Forssk.) DC. | Per. | Ch. | SA-AR + SU-ZA |
| | [#] *Kleinia pendula* DC. | Per. | Ch. | SA-AR + SU-ZA |
| | *Launaea intybacea* (Jacq.) Beauverd | Ann. | Th. | SA-AR + SU-ZA |
| | *Launaea massauensis* (Fresen.) Sch.Bip. ex Kuntze | Ann. | Th. | IR-TR + ME + SA-AR + SU-ZA |
| | *Microglossa pyrrhopappa* (A.Rich.) Agnew | Per. | Ph. | SA-AR + SU-ZA |
| | * *Onopordum heteracanthum* C.A.Mey. | Per. | Ch. | IR-TR + ME + SA-AR |
| | * *Picris scabra* Forssk. | Per. | Ch. | SA-AR |
| | *Psiadia punctulata* Vatke | Per. | Ch. | SA-AR + SU-ZA |
| | *Pulicaria petiolaris* Jaub. & Spach | Per. | Ch. | IR-TR + SA-AR + SU-ZA |
| | *Pulicaria undulata* (L.) C.A.Mey. | Per. | He. | IR-TR + SA-SI + SU-ZA |
| | *Pulicaria schimperi* DC. | Per. | Ch. | SA-AR + SU-ZA |
| | *Reichardia tingitana* (L.) Roth | Ann. | Th. | IR-TR + ME + SA-AR + SU-ZA |
| | *Senecio hadiensis* Forssk. | Per. | He. | SA-AR + SU-ZA |
| | [#] *Solanecio angulatus* (Vahl) C.Jeffrey | Per. | He. | SA-AR + SU-ZA |

| Family | Taxa | Life Span | Life Form | Chorotype |
|---|---|---|---|---|
| | *Sonchus oleraceus* L. | Ann. | Th. | COSM |
| | ‡ *Tagetes minuta* L. | Ann. | Th. | NEO |
| | ‡ *Tridax procumbens* L. | Per. | Ch. | NEO |
| | *,‡ *Verbesina encelioides* (Cav.) Benth. & Hook.f. ex A.Gray | Ann. | Th. | NEO |
| | * *Xanthium strumarium* L. | Ann. | Th. | COSM |
| **Boraginaceae** | *Alkanna orientalis* (L.) Boiss. | Per. | Ch. | IR-TR + ME + SA-AR + SU-ZA |
| | *Arnebia hispidissima* (Sieber ex Lehm.) A.DC. | Ann. | Th. | IR-TR + SA-AR + SU-ZA |
| | * *Cordia monoica* Roxb. | Per. | Ph. | PAL |
| | * *Cordia sinensis* Lam. | Per. | Ph. | PAL |
| | *Cynoglossum bottae* Deflers | Per. | Ph. | SA-AR |
| | *Ehretia cymosa* Thonn. | Per. | Ph. | SA-AR + SU-ZA |
| | # *Ehretia obtusifolia* Hochst. ex A.DC. | Per. | Ph. | PAL |
| | * *Heliotropium arbainense* Fresen. | Per. | Ch. | IR-TR + ME + SA-AR + SU-ZA |
| | *Heliotropium longiflorum* (A.DC.) Jaub. & Spach | Per. | Ch. | SA-AR + SU-ZA |
| | *Heliotropium zeylanicum* (Burm.f.) Lam. | Per. | Ch. | SA-AR + SU-ZA |
| **Brassicaceae** | *Erucastrum arabicum* Fisch. & C.A.Mey. | Ann. | Th. | SA-AR + SU-ZA |
| | *Sisymbrium erysimoides* Desf. | Ann. | Th. | ME + SA-SI + SU-ZA |
| | *Sisymbrium irio* L. | Ann. | Th. | IR-TR + ME + SA-AR + SU-ZA |
| **Burseraceae** | * *Commiphora gileadensis* (L.) C.Chr. | Per. | Ph. | SA-AR + SU-ZA |
| | # *Commiphora kataf* (Forssk.) Engl. | Per. | Ph. | SA-AR + SU-ZA |
| | *Commiphora kua* (R.Br. ex Royle) Vollesen | Per. | Ph. | SA-AR + SU-ZA |
| **Cactaceae** | ‡ *Opuntia dillenii* (Ker Gawl.) Haw. | Per. | Ph. | NEO |
| | ‡ *Opuntia ficus-indica* (L.) Mill. | Per. | Ph. | NEO |
| **Campanulaceae** | *Campanula edulis* Forssk. | Per. | He. | SA-AR + SU-ZA |
| **Capparaceae** | * *Boscia integrifolia* J.St.-Hil. | Per. | Ph. | SA-AR + SU-ZA |
| | * *Cadaba glandulosa* Forssk. | Per. | Ph. | IR-TR + SA-AR + SU-ZA |
| | *Capparis spinosa* var. *aegyptia* (Lam.) Boiss. | Per. | Ph. | IR-TR + ME + SA-SI |
| **Caryophyllaceae** | *,# *Gypsophila umbricola* (J.R.I.Wood) R.A.Clement | Per. | Ch. | SA-AR |
| | *Polycarpon tetraphyllum* (L.) L. | Ann. | Th. | COSM |
| | * *Silene burchellii* Otth | Ann. | Th. | SU-ZA |
| **Celastraceae** | *Catha edulis* (Vahl) Endl. | Per. | Ph. | SU-ZA |
| | * *Gymnosporia parviflora* (Vahl) Chiov. | Per. | Ph. | SA-AR |
| | *Gymnosporia senegalensis* (Lam.) Loes. | Per. | Ph. | SA-AR + SU-ZA |
| **Cleomaceae** | * *Cleome brachycarpa* Vahl ex DC. | Ann. | Th. | IR-TR + SA-SI + SU-ZA |
| | *Cleome gynandra* L. | Ann. | Th. | PAN |
| | * *Cleome paradoxa* R.Br. ex DC. | Per. | Ch. | SA-AR + SU-ZA |
| | * *Cleome ramosissima* Parl. ex Webb | Ann. | Th. | SA-AR + SU-ZA |
| **Combretaceae** | # *Combretum aculeatum* Vent. | Per. | Ph. | SA-AR + SU-ZA |
| | * *Combretum molle* R.Br. ex G.Don | Per. | Ph. | SA-AR + SU-ZA |
| | *Terminalia brownii* Fresen. | Per. | Ph. | SA-AR + SU-ZA |
| **Commelinaceae** | * *Aneilema forskalii* Kunth | Ann. | Th. | SA-AR + SU-ZA |
| | * *Commelina albescens* Hassk. | Per. | He. | IR-TR + SA-AR + SU-ZA |
| | *Commelina benghalensis* L. | Per. | He. | PAL |
| | *Commelina forskaolii* Vahl | Per. | Ch. | PAL |
| | *Cyanotis nyctitropa* Deflers | Per. | Ch. | SA-AR |

| Family | Taxa | Life Span | Life Form | Chorotype |
|---|---|---|---|---|
| **Convolvulaceae** | *Evolvulus alsinoides* (L.) L. | Per. | Ch. | PAN |
| | *Ipomoea obscura* (L.) Ker Gawl. | Per. | Ch. | PAL |
| **Crassulaceae** | *Crassula schimperi* Fisch. & C.A.Mey. | Per. | Ch. | IR-TR + SA-AR + SU-ZA |
| | *Kalanchoe crenata* (Andrews) Haw. | Per. | Ph. | PAL |
| | *Kalanchoe glaucescens* Britten | Per. | Ph. | SA-AR + SU-ZA |
| | # *Kalanchoe laciniata* (L.) DC. | Per. | Ph. | SU-ZA |
| **Cucurbitaceae** | *Citrullus colocynthis* (L.) Schrad. | Per. | He. | IR-TR + ME + SA-SI + SU-ZA |
| | *Coccinia grandis* (L.) Voigt | Per. | He. | PAL |
| | *Cucumis melo* L. | Ann. | Th. | PAN |
| | *Cucumis prophetarum* L. | Per. | He. | IR-TR + ME + SA-SI +SU-ZA |
| | *Kedrostis foetidissima* (Jacq.) Cogn. | Per. | Ph. | SA-AR + SU-ZA |
| | * *Kedrostis gijef* (Forssk. ex J.F.Gmel.) C.Jeffrey | Per. | Ph. | SA-AR + SU-ZA |
| | *Momordica balsamina* L. | Ann. | Th. | SA-AR + SU-ZA |
| **Cupressacceae** | *Juniperus procera* Hochst. ex Endl. | Per. | Ph. | SA-AR + SU-ZA |
| **Cyperaceae** | *,# *Cyperus alternifolius* subsp. *flabelliformis* Kük. | Per. | GH | SA-AR + SU-ZA |
| | * *Cyperus cruentus* Rottb. | Per. | He. | SA-AR +SU-ZA |
| | *Cyperus niveus* var. *leucocephalus* (Kunth) Fosberg | Per. | He. | SA-AR +SU-ZA |
| | *Cyperus rubicundus* Vahl | Ann. | Th. | PAN |
| | * *Schoenus nigricans* L. | Per. | He. | COSM |
| **Euphorbiaceae** | *Acalypha fruticosa* Forssk. | Per. | Ch. | PAL |
| | *Acalypha paniculata* Miq. | Per. | Ch. | PAL |
| | *Chrozophora oblongifolia* (Delile) A.Juss. ex Spreng. | Per. | Ch. | SA-SI + SU-ZA |
| | # *Euphorbia ammak* Schweinf. | Per. | Ph. | SA-AR |
| | * *Euphorbia arabica* Hochst. & Steud. ex T.Anderson | Per. | He. | ME + SA-AR + SU-ZA |
| | *Euphorbia cactus* Ehrenb. ex Boiss. | Per. | Ph. | SA-AR + SU-ZA |
| | * *Euphorbia fractiflexa* S.Carter & J.R.I.Wood | Per. | Ph. | SA-AR |
| | *Euphorbia granulata* Forssk. | Ann. | Th. | IR-TR + SA-SI + SU-ZA |
| | ‡ *Euphorbia hirta* L. | Ann. | Th. | NEO |
| | *Euphorbia inarticulata* Schweinf. | Per. | Ch. | SA-AR + SU-ZA |
| | *Euphorbia schimperiana* Scheele | Per. | He. | SA-AR + SU-ZA |
| | ‡ *Jatropha curcas* L. | Per. | Ph. | NEO |
| | *Ricinus communis* L. | Per. | Ph. | SU-ZA |
| | *Tragia pungens* (Forssk.) Müll.Arg. | Per. | Ch. | SA-AR + SU-ZA |
| **Fabaceae** | *Abrus bottae* Deflers | Per. | Ph. | SA-AR |
| | # *Abrus precatorius* L. | Per. | Ph. | PAN |
| | *Argyrolobium arabicum* (Decne.) Jaub. & Spach | Ann. | Th. | SA-SI + SU-ZA |
| | *Astragalus atropilosulus* (Hochst.) Bunge | Per. | Ch. | SU-ZA |
| | * *Cadia purpurea* (G.Piccioli) Aiton | Per. | Ph. | SA-AR + SU-ZA |
| | * *Canavalia cathartica* Thouars | Per. | Ph. | PAN |
| | *Crotalaria incana* L. | Ann. | Th. | NEO |
| | * *Crotalaria retusa* L. | Ann. | Th. | PAN |
| | *Delonix elata* (L.) Gamble | Per. | Ph. | SA-AR + SU-ZA |
| | *Dichrostachys cinerea* (L.) Wight & Arn. | Per. | Ph. | DEC |
| | *Dolichos trilobus* L. | Per. | He. | PAL |
| | *Dorycnopsis abyssinica* (A.Rich.) V.N.Tikhom. & Sokoloff | Per. | He. | SU-ZA |
| | *Indigofera articulata* Gouan | Per. | Ph. | IR-TR + SA-AR + SU-ZA |

| Family | Taxa | Life Span | Life Form | Chorotype |
|---|---|---|---|---|
| | *Indigofera coerulea* Roxb. | Per. | Ch. | SA-SI + SU-ZA |
| | *Indigofera hochstetteri* Baker | Ann. | Th. | SA-SI + SU-ZA |
| | *Indigofera oblongifolia* Forssk. | Per. | Ch. | SA-SI + SU-ZA |
| | *Indigofera spinosa* Forssk. | Per. | Ch. | SA-AR + SU-ZA |
| | *Indigofera tinctoria* L. | Per. | Ph. | PAN |
| | *Lablab purpureus* (L.) Sweet | Per. | Ch. | SU-ZA |
| | * *Microcharis tritoides* (Baker) Schrire | Per. | Ph. | SA-AR + SU-ZA |
| | *Neonotonia wightii* (Wight & Arn.) J.A.Lackey | Per. | He. | SU-ZA |
| | * *Parkinsonia aculeata* L. | Per. | Ph. | NEO |
| | # *Pterolobium stellatum* (Forssk.) Brenan | Per. | Ph. | SU-ZA |
| | * *Rhynchosia minima* (L.) DC. | Per. | Ch. | PAN |
| | *Senegalia asak* (Forssk.) Kyal. & Boatwr. | Per. | Ph. | SA-AR + SU-ZA |
| | * *Senegalia laeta* (R.Br. ex Benth.) Seigler & Ebinger | Per. | Ph. | SA-AR + SU-ZA |
| | *Senegalia mellifera* (Benth.) Seigler & Ebinger | Per. | Ph. | SA-AR + SU-ZA |
| | *Senna alexandrina* Mill. | Per. | He. | SA-SI + SU-ZA |
| | * *Senna italica* Mill. | Per. | Ch. | IR-TR + ME + SA-SI + SU-ZA |
| | *Senna occidentalis* (L.) Link | Per. | Ch. | NEO |
| | * *Sesbania leptocarpa* DC. | Per. | Ph. | SU-ZA |
| | *,# *Sesbania sericea* (Willd.) Link | Per. | Ph. | SA-AR + SU-ZA |
| | *Tamarindus indica* L. | Per. | Ph. | SU-ZA |
| | *Vachellia etbaica* (Schweinf.) Kyal. & Boatwr. | Per. | Ph. | SA-AR + SU-ZA |
| | * *Vachellia johnwoodii* (Boulos) Ragup., Seigler, Ebinger & Maslin | Per. | Ph. | SA-AR |
| | # *Vachellia seyal* (Delile) P.J.H.Hurter | Per. | Ph. | SA-AR + SU-ZA |
| | *Vachellia tortilis* (Forssk.) Galasso & Banfi | Per. | Ph. | IR-TR + ME + SA-AR + SU-ZA |
| | *Vigna membranacea* A.Rich. | Ann. | Th. | SU-ZA |
| **Gentianaceae** | * *Enicostema axillare* (Poir. ex Lam.) A.Raynal | Per. | Ph. | PAL |
| **Geraniaceae** | * *Geranium trilophum* Boiss. | Ann. | Th. | IR-TR + SA-AR + SU-ZA |
| | *Pelargonium multibracteatum* Hochst. Ex A.Rich. | Per. | He. | SA-AR + SU-ZA |
| **Gisekiaceae** | *Gisekia pharnaceoides* L. | Ann. | Th. | PAL |
| **Lamiaceae** | *Coleus arabicus* Benth. | Per. | Ch. | SA-AR |
| | * *Coleus barbatus* (Andrews) Benth. ex G.Don | Per. | Ch. | PAL |
| | * *Endostemon tenuiflorus* (Benth.) M.R.Ashby | Ann. | Th. | SA-AR + SU-ZA |
| | *Lavandula coronopifolia* Poir. | Per. | Ch. | IR-TR + ME + SA-AR + SU-ZA |
| | *Lavandula pubescens* Decne. | Per. | Ch. | ME + SA-AR + SU-ZA |
| | *Leucas alba* (Forssk.) Sebald | Per. | Ch. | SA-AR |
| | *Micromeria imbricata* (Forssk.) C.Chr. | Per. | Ch. | SA-AR + SU-ZA |
| | *Nepeta deflersiana* Schweinf. ex Hedge | Per. | He. | SA-AR |
| | *Ocimum filamentosum* Forssk. | Per. | Ph. | PAL |
| | *Ocimum forskaolii* Benth. | Per. | Ch. | SA-AR + SU-ZA |
| | *Ocimum serpyllifolium* Forssk. | Per. | Ch. | SA-AR + SU-ZA |
| | *Orthosiphon pallidus* Royle ex Benth. | Per. | Ch. | SA-SI + SU-ZA |
| | *Otostegia fruticosa* (Forssk.) Schweinf. ex Penzig | Per. | Ph. | ME + SA-AR + SU-ZA |
| | *Teucrium yemense* Deflers | Per. | Ch. | SA-AR + SU-ZA |
| **Linderniaceae** | * *Craterostigma plantagineum* Hochst. | Per. | He. | SU-ZA |
| | *Craterostigma pumilum* Hochst. | Per. | He. | SA-AR + SU-ZA |

| Family | Taxa | Life Span | Life Form | Chorotype |
|---|---|---|---|---|
| **Lophiocarpaceae** | *Corbichonia decumbens* (Forssk.) Exell | Ann. | Th. | IR-TR + SA-SI + SU-ZA |
| **Loranthaceae** | * *Plicosepalus curviflorus* (Benth. ex Oliv.) Tiegh. | Per. | Pa. | SA-AR + SU-ZA |
| | *Tapinanthus globifer* (A.Rich.) Tiegh. | Per. | Pa. | SA-AR + SU-ZA |
| **Lythraceae** | *Lawsonia inermis* L. | Per. | Ph. | PAL |
| **Malvaceae** | *Abutilon bidentatum* Hochst. ex A.Rich. | Per. | Ch. | PAL |
| | *Abutilon pannosum* (G.Forst.) Schltdl. | Per. | Ch. | IR-TR + ME + SA-SI + SU-ZA |
| | *Grewia tembensis* Fresen. | Per. | Ph. | SA-AR + SU-ZA |
| | *Grewia trichocarpa* Hochst. ex A.Rich. | Per. | Ph. | SA-AR + SU-ZA |
| | *Grewia velutina* (Forssk.) Vahl | Per. | Ph. | SA-AR + SU-ZA |
| | *Hibiscus aponeurus* Sprague & Hutch. | Per. | Ch. | SU-ZA |
| | *Hibiscus deflersii* Schweinf. ex Cufod. | Per. | Ch. | SA-AR + SU-ZA |
| | *Hibiscus palmatus* Forssk. | Per. | Ch. | IR-TR + SA-SI + SU-ZA |
| | * *Malva parviflora* L. | Ann. | Th. | IR-TR + ME + SA-SI |
| | *Melhania incana* B.Heyne ex Wight & Arn. | Per. | Ch. | SA-AR + SU-ZA |
| | *Pavonia burchellii* (DC.) R.A.Dyer | Per. | Ch. | SA-AR + SU-ZA |
| | *Sida ovata* Forssk | Ann. | Th. | IR-TR + SA-SI + SU-ZA |
| | *Triumfetta rhomboidea* Jacq. | Ann. | Th. | PAN |
| **Meliaceae** | *Trichilia emetica* Vahl | Per. | Ph. | SA-AR + SU-ZA |
| **Menispermaceae** | * *Cocculus pendulus* (J.R.Forst. & G.Forst.) Diels | Per. | Ph. | IR-TR + ME + SA-SI + SU-ZA |
| **Molluginaceae** | *Hypertelis cerviana* (L.) Thulin | Ann. | Th. | COSM |
| | * *Paramollugo nudicaulis* (Lam.)Thulin | Ann. | Th. | PAL |
| **Moraceae** | *Ficus salicifolia* Vahl | Per. | Ph. | SA-AR + SU-ZA |
| | * *Ficus glumosa* Delile | Per. | Ph. | SA-AR + SU-ZA |
| | *Ficus ingens* (Miq.) Miq. | Per. | Ph. | SA-AR + SU-ZA |
| | *Ficus palmata* subsp. *virgata* Browicz | Per. | Ph. | ME + SA-AR + SU-ZA |
| | *,# *Ficus populifolia* Vahl | Per. | Ph. | SA-AR + SU-ZA |
| | *Ficus sycomorus* L. | Per. | Ph. | ME + SA-AR + SU-ZA |
| | *Ficus vasta* Forssk. | Per. | Ph. | SA-AR + SU-ZA |
| **Nyctaginaceae** | *Boerhavia diffusa* L. | Per. | Ch. | COSM |
| | *Boerhavia repens* L. | Per. | Ch. | PAN |
| | # *Commicarpus ambiguus* Meikle | Per. | Ch. | SA-AR + SU-ZA |
| | *Commicarpus grandiflorus* (A.Rich.) Standl. | Per. | Ch. | SA-SI + SU-ZA |
| | *Commicarpus plumbagineus* (Cav.) Standl. | Per. | Ch. | IR-TR + ME + SA-SI + SU-ZA |
| **Oleaceae** | *Jasminum grandiflorum* subsp. *floribundum* (R.Br. ex Fresen.) P.S.Green | Per. | Ph. | SA-AR + SU-ZA |
| | *Olea europaea* L. | Per. | Ph. | IR-TR + ME + SA-AR + SU-ZA |
| **Orobanchaceae** | * *Cistanche phelypaea* (L.) Cout. | Per. | Pa. | IR-TR + ME + SA-AR + SU-ZA |
| | *Orobanche minor* Sm. | Ann. | Pa. | IR-TR + ME + SA-AR + SU-ZA |
| **Oxalidaceae** | *Oxalis corniculata* L. | Ann. | Th. | NEO |
| **Papaveraceae** | ‡ *Argemone mexicana* L. | Ann. | Th. | PAN |
| | *,‡ *Argemone ochroleuca* Sweet | Ann. | Th. | NEO |
| | *,# *Fumaria abyssinica* Hammar | Ann. | Th. | SA-AR + SU-ZA |
| **Passifloraceae** | *,# *Adenia venenata* Forssk. | Per. | Ph. | SA-AR + SU-ZA |

| Family | Taxa | Life Span | Life Form | Chorotype |
|---|---|---|---|---|
| **Peraceae** | *Clutia lanceolata* Forssk. | Per. | Ph. | SA-AR + SU-ZA |
| **Plantaginaceae** | * *Kickxia petiolata* D.A.Sutton | Per. | Ch. | SU-ZA |
| | *Schweinfurthia pterosperma* (A.Rich.) A.Braun | Ann. | Th. | IR-TR + SA-AR + SU-ZA |
| **Plumbaginaceae** | # *Plumbago zeylanica* L. | Per. | Ch. | PAN |
| **Poaceae** | * *Aristida congesta* Roem. & Schult. | Ann. | Th. | IR-TR + ME + SA-AR + SU-ZA |
| | * *Cenchrus biflorus* Roxb. | Ann. | Th. | SA-SI + SU-ZA |
| | * *Cenchrus longisetus* M.C.Johnst. | Per. | He. | SA-AR + SU-ZA |
| | *Cenchrus setaceus* (Forssk.) Morrone | Per. | He. | IR-TR + ME + SA-AR + SU-ZA |
| | ‡ *Cenchrus setigerus* Vahl | Ann. | Th. | PAL |
| | * *Chloris flagellifera* (Nees) P.M.Peterson | Per. | He. | IR-TR + SA-AR + SU-ZA |
| | *Chloris gayana* Kunth | Ann. | Th. | SA-AR + SU-ZA |
| | * *Chrysopogon plumulosus* Hochst. | Per. | He. | SA-AR + SU-ZA |
| | * *Dactyloctenium scindicum* Boiss. | Ann. | Th. | IR-TR + SA-AR + SU-ZA |
| | * *Danthoniopsis barbata* (Nees) C.E.Hubb. | Per. | He. | SA-AR + SU-ZA |
| | *Digitaria ciliaris* (Retz.) Koeler | Ann. | Th. | PAL |
| | *Digitaria nodosa* Parl. | Per. | He. | IR-TR + ME + SA-AR + SU-ZA |
| | *Digitaria velutina* (Forssk.) P.Beauv. | Ann. | Th. | SA-AR + SU-ZA |
| | * *Enneapogon cenchroides* (Licht.) C.E.Hubb. | Ann. | Th. | SA-AR + SU-ZA |
| | *Enneapogon lophotrichus* Chiov. ex H.Scholz & P.König | Ann. | Th. | SA-AR + SU-ZA |
| | *Eragrostis barrelieri* Daveau | Ann. | Th. | IR-TR + ME + SA-AR + SU-ZA |
| | *Eragrostis papposa* (Roem. & Schult.) Steud. | Per. | He. | IR-TR + ME + SA-AR + SU-ZA |
| | *Hyparrhenia hirta* (L.) Stapf | Per. | He. | IR-TR + ME + SA-AR + SU-ZA |
| | * *Leptothrium senegalense* (Kunth) Clayton | Per. | He. | IR-TR + SA-AR + SU-ZA |
| | * *Megathyrsus maximus* (Jacq.) B.K.Simon & S.W.L.Jacobs | Per. | He. | SA-AR + SU-ZA |
| | *Melinis repens* (Willd.) Zizka | Ann. | Th. | ME + SA-AR + SU-ZA |
| | * *Paspalum vaginatum* Sw. | Per. | He. | NEO |
| | * *Schoenefeldia gracilis* Kunth | Ann. | Th. | SA-SI + SU-ZA |
| | *Sehima nervosum* (Rottler) Stapf | Per. | He. | PAL |
| | * *Sporobolus ioclados* (Nees ex Trin.) Nees | Per. | He. | IR-TR + ME + SA-SI + SU-ZA |
| | *Stipagrostis hirtigluma* (Steud. ex Trin. & Rupr.) De Winter | Ann. | Th. | IR-TR + SA-AR + SU-ZA |
| | * *Tetrapogon cenchriformis* (A.Rich.) Clayton | Ann. | Th. | SA-AR + SU-ZA |
| | * *Tetrapogon tenellus* (J.Koenig ex Roxb.) Chiov. | Per. | He. | SA-SI + SU-ZA |
| | *Themeda triandra* Forssk. | Per. | He. | PAN |
| | *Tricholaena teneriffae* (L.f.) Link | Per. | He. | IR-TR + ME + SA-SI + SU-ZA |
| **Polygalaceae** | *Polygala erioptera* DC. | Ann. | Th. | PAL |
| | *Polygala tinctoria* Vahl | Ann. | Th. | SA-AR + SU-ZA |
| **Polygonaceae** | *Oxygonum sinuatum* (Hochst. & Steud. ex Meisn.) Dammer | Ann. | Th. | SA-AR + SU-ZA |
| | *Rumex nervosus* Vahl | Ann. | Th. | SA-AR + SU-ZA |
| **Portulacaeae** | *Portulaca oleracea* L. | Ann. | Th. | IR-TR + ME + SA-AR + SU-ZA |

| Family | Taxa | Life Span | Life Form | Chorotype |
|---|---|---|---|---|
| **Primulaceae** | *Lysimachia arvensis* (L.) U.Manns & Anderb. | Ann. | Th. | COSM |
| **Resedaceae** | *Ochradenus baccatus* Delile | Per. | Ph. | IR-TR + SA-AR + SU-ZA |
| | + *Reseda sphenocleoides* Deflers | Per. | Ch. | SA-AR + SU-ZA |
| **Rhamnaceae** | *,# *Ziziphus mucronata* Willd. | Per. | Ph. | SA-AR + SU-ZA |
| | *Ziziphus spina-christi* (L.) Desf. | Per. | Ph. | IR-TR + ME + SA-AR + SU-ZA |
| | *Berchemia discolor* (Klotzsch) Hemsl. | Per. | Ph. | SA-AR + SU-ZA |
| **Rubiaceae** | *Oldenlandia capensis* L.f. | Ann. | Th. | IR-TR + ME + SA-AR + SU-ZA |
| | *Pavetta longiflora* Vahl | Per. | Ph. | SA-AR + SU-ZA |
| | *Pyrostria phyllanthoidea* (Baill.) Bridson | Per. | Ph. | SA-AR + SU-ZA |
| **Salvadoraceae** | *Dobera glabra* (Forssk.) Juss. ex Poir. | Per. | Ph. | SA-AR + SU-ZA |
| | *Salvadora persica* L. | Per. | Ph. | IR-TR + ME + SA-AR + SU-ZA |
| **Sapindaceae** | *Dodonaea viscosa* subsp. *angustifolia* (L.f.) J.G.West | Per. | Ph. | PAN |
| **Scrophulariaceae** | * *Anticharis senegalensis* (Walp.) Bhandari | Ann. | Th. | SA-SI + SU-ZA |
| | * *Buddleja polystachya* Fresen. | Per. | Ph. | SA-AR + SU-ZA |
| | *Rhabdotosperma bottae* (Deflers) Hartl | Per. | Ch. | SA-AR + SU-ZA |
| | * *Scrophularia arguta* Aiton | Ann. | Th. | IR-TR + ME + SA-AR + SU-ZA |
| | * *Verbascum asiricum* Hemaid | Ann. | Th. | SA-AR |
| **Solanaceae** | *,‡ *Datura innoxia* Mill. | Ann. | Th. | NEO |
| | *Nicotiana tabacum* L. | Ann. | Th. | NEO |
| | *Solanum incanum* L. | Per. | Ch. | PAL |
| | *Solanum schimperianum* Hochst. | Per. | Ch. | SA-AR + SU-ZA |
| | *Solanum virginianum* L. | Ann. | Th. | PAL |
| | *Solanum villosum* Mill. | Ann. | Th. | COSM |
| | *Withania somnifera* (L.) Dunal | Per. | Ch. | IR-TR + ME + SA-SI + SU-ZA |
| **Talinaceae** | *Talinum portulacifolium* (Forssk.) Asch. ex Schweinf. | Per. | Ch. | SA-AR + SU-ZA |
| **Tamaricaceae** | *Tamarix nilotica* (Ehrenb.) Bunge | Per. | Ph. | ME + SA-AR + SU-ZA |
| | * *Tamarix aphylla* (L.) H.Karst. | Per. | Ph. | IR-TR + SA-SI + SU-ZA |
| **Thymelaeaceae** | * *Lasiosiphon somalensis* (Franch.) H.Pearson | Per. | Ch. | SA-AR + SU-ZA |
| **Urticaceae** | *Urtica urens* L. | Ann. | Th. | COSM |
| | *Forsskaolea tenacissima* L. | Per. | Ch. | IR-TR + ME + SA-SI + SU-ZA |
| **Verbenaceae** | *Chascanum marrubiifolium* Fenzl ex Walp. | Per. | Ch. | SA-SI + SU-ZA |
| | ‡ *Lantana camara* L. | Per. | Ph. | NEO |
| | *Phyla nodiflora* (L.) Greene | Per. | He. | PAN |
| | *Priva cordifolia* (L.f.) Druce | Per. | Ch. | PAL |
| **Vitaceae** | *Cissus rotundifolia* Vahl | Per. | He. | SA-AR + SU-ZA |
| | *Cissus quadrangularis* L. | Per. | Ph. | PAL |
| | *Cyphostemma digitatum* (Forssk.) Desc. | Per. | Ch. | SA-AR + SU-ZA |
| | *Rhoicissus revoilii* Planch. | Per. | Ph. | SA-AR + SU-ZA |
| **Zygophyllaceae** | *,# *Balanites aegyptiaca* (L.) Delile | Per. | Ph. | IR-TR + ME + SA-AR + SU-ZA |
| | * *Tribulus parvispinus* C.Presl | Ann. | Th. | IR-TR + SA-AR + SU-ZA |

Legend: *: new records, +: endemic not endangered, #: non-endemic-endangered, ^: endemic-endangered, ‡: Exotic species. Chorotypes abbreviations (see Table 2). Life span: Ann. Annual, Per.: Perennial. Life form: Ch.: Chamaephyte, Ge.: Geophyte, GH: Geophyte-Helophyte, He.: Hemicryptophyte, Pa.: Parasite, Ph.: Phanerophyte, Th.: Therophyte.

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
