# Peer review of "Floristic Diversity and Phytogeography of JABAL Fayfa: A Subtropical Dry Zone, South-West Saudi Arabia"

_diversity, doi:10.3390/d12090345_

Round 1

Reviewer 1 Report

This paper describes the flora of the Jebel Fayfa region and will be an essential piece of the knowledge of the biological diversity of this region. Therefore, I believe that it is worthy to be published. After that, many researchers refer this paper in the future. For the future readers, however, authors should provide more detail information about each description. I listed detailed comments below. I hope that these help you revise your manuscript.

Line 48 “from seas level to 9,300 meters”,

Really 9,300 meters? It is higher than Mt. Everest.

Figure 1 and line 84 “16 localities”,

I think that authors must chose these localities thoughtfully as explained line 85-87. If more detailed descriptions of each localities are provided (for example, as a table), it helps readers’ understanding.

Table 1 acronyms such as “ME + SA–AR”,

The explanations of the acronyms are provided as the footnote of Appendix A. However, appendix material are supplemental, not main contents. The explanations should be in the main contents (as text or footnote of the table).

Figure 3, 4 and 5,

These figures are not informative. The same information provided in these figures are provided in the text. Authors must have got vast data through a series of field surveys but they presented pooled list. I think authors can provide more detailed analysis. For example, seasonal or distributional data may be presented.

Line 133 “Chorological analysis”,

What literatures were cited for the analysis? What is the goal this analysis? Authors should explicitly explain them. For example, I am not familiar with the flora of Middle East and the categories used in this analysis.

Author Response

Response to Reviewer 1 Comments

Point 1: Line 48 “from seas level to 9,300 meters”, Really 9,300 meters? It is higher than Mt. Everest.

 Response 1: ‘9,300 meters’ replaced with ‘3,100 meters’.

Point 2: Figure 1 and line 84 “16 localities”, I think that authors must chose these localities thoughtfully as explained line 85-87. If more detailed descriptions of each localities are provided (for example, as a table), it helps readers’ understanding.

Response 2: the localities are presented in Table 1.

Point 3: Table 1 acronyms such as “ME + SA–AR”, The explanations of the acronyms are provided as the footnote of Appendix A. However, appendix material is supplemental, not the main contents. The explanations should be in the main contents (as text or footnote of the table).

Response 3: The number of the previous Table 1 changed into Table 2

‘The explanations of Table 2 acronyms are provided as a footnote of the table’.

Point 4: “Figure 3, 4 and 5”, These figures are not informative. The same information provided in these figures are provided in the text. Authors must have got vast data through a series of field surveys, but they presented pooled list. I think authors can provide more detailed analysis. For example, seasonal or distributional data may be presented.

Response 4: ‘Actually this is a good suggestion, but the seasonal or distributional data will be presented in a new paper about the study area’.

Point 5: Line 133 “Chorological analysis”, What literatures were cited for the analysis? What is the goal of this analysis? The authors should explicitly explain them. For example, I am not familiar with the flora of the Middle East and the categories used in this analysis.

Response 5: The literatures were cited for the analysis on line 90 in the Materials and Methods section as [10,49–52].

Chorological analysis aims to study the spatial distribution of organisms (biogeography). It refers to the study of the geographical and topographical spread of organisms away from a center of origin. Chorological categories indicate the study aimed at the identification of recurrent patterns. In the geography of floras, certain patterns of distribution frequently recur, bringing the plants they represent into geographical relationships. Distributions that are centered in the same region and whose boundaries broadly overlap can be grouped together and regarded as members of the same chorological category.

Reviewer 2 Report

The paper "Floristic diversity and phytogeography of Jabal Fayfa, Jizan Province, South-West Saudi Arabia" represent a useful contribution to the knowledge of the plant diversity of a sector in the Arabian peninsula not well know until now.

The paper is well written and results and discussion are clearly presented.

I recommend to publish this paper with minor revisions as specified below:

- to provide in the text the informations regarding mean annual temperature and precipitation over a longer time series (at least 20 years, if available) for the cited meteorological station;

- to indicate the altitude of this station;

- to include in results and discussion data and comments about the presence of endemic species (are there some taxa endemic of the study area or not?; did you have found some taxa endemic of Saudi Arabia or not? How many?

Are there, in the vascular flora of the study area, some species judged particularly rare in Saudi Arabia or considered endangered of extinction? Please, discuss this issue in the text.

How many alien taxa are considered present in the study area? Are they invasive? Can they threaten the native flora or local ecosystems? Please discuss this topic.

I suggest to add an appendix with photographs of relevant taxa and landscapes found in the study area.

Other annotations can be found in the annotated pdf file in attachment.

Author Response

Response to Reviewer 2 Comments

Point 1: Line 16, please check this sentence “ four elements of the of vegetation”

 Response 1:  An extra word ‘of ’ is deleted.

Point 2: 2.5% of endemic species doesn't appear to me as a "large" number! If you want retain this you have to compare this data with percentage of endemic taxa found in the other countries of this region.

Response 2: Line 39, the word ‘a large’ is deleted

Point 3: Do you mean: it is important because it represents a link between the two continents? If yes, please reword to explicit the concept

Response 3: Line 45, the sentence ‘because of its important position with regard to the continents of Asia and Africa’ reword into ‘because it represents a link between the two Asia and Africa’.

Point 4: Line 48, please check the number! this is meters, not feet.

Response 4: Line 48:‘9,300 meters’ replaced with ‘3,100 meters’ 

Point 5: Line 50, need reference.

Response 5: Line 50: the reference ‘ [2] ’ is added.

Point 6: Line 75, These are interesting information, however it is better to provide also mean values of temperature and rainfall considering a series of at least 20 years of meteorological records; if not available you could retrieve data from some of the world climate dataset, e.g. Worldclim...

Response 6: Line 75: Metrological data maximum temperature, minimum temperature, and precipitation during the last 20 years were put and (Figure 2) was updated.

Point 7: Line 77, please specify the altitude of the Station.

Response 7: Line 77: altitude of Jizan City Meteorological Station 16°53'48.5"N 42°35'02.4"E

Point 8: Line 77, please specify the altitude of the Station.

Response 8: Line 77, altitude of the Station is 16°53'48.5"N 42°35'02.4"E

Point 9: Line 85, It is not clear what do you mean with"stands": they are vegetation plot? they are of equal size and shape? they are not precisely defined in the field. Please specify.

Response 9: line 85, ‘Stand (a vegetation plot)’, it is an aggregation of plants occupying a specific area and sufficiently uniform in species composition, size, age, arrangement, and condition.

Point 10: line 86, Please specify how do you performed the selection of stands

Response 10: line 86, ‘In each location, sampling stands were situated randomly using the réléve method [38]’.

Point 11: line 125, lifeforms.

Response 11: line 125, ‘Lifeforms’ replaced by ‘life forms’.

Point 12:line 126, lifeform.

Response 12:line 126, ‘Lifeform’ replaced by ‘life form’.

Point 13: line 156, insert space.

Response 13: space is inserted between the chorotypes.

Point 14:line 214, chaemophytes ???

Response 14:line 214, ‘chaemophytes’ replaced with ‘chamaephytes’

Point 15: line 305, ‘contribution’, Capital letter

Response 15: line 305, ‘Contribution’

Point 16:line 365, missed reference.

Response 16: line 365, ‘38.   Muller-Dombois, D.; Ellenberg, H. Aims and methods of vegetation ecology. John Wiley and Sons; New York, 1974; pp. 547’ added.

Point 17: to include in results and discussion data and comments about the presence of endemic species (are there some taxa endemic of the study area or not?; did you have found some taxa endemic of Saudi Arabia or not? How many?

Response 17:There are six endemic taxa in the study area, of them five are endemic not endangered (Anisotes trisulcus, Barleria bispinosa, Barleria trispinosa, Ceropegia aristolochioides, and Reseda sphenocleoides), and one is endemic endangered (Aloe woodii). The same taxa are endemic to Saudi Arabia.

Point 18: Are there, in the vascular flora of the study area, some species judged particularly rare in Saudi Arabia or considered endangered or extinction? Please, discuss this issue in the text

Response 18: Yes, there are 27 species considered endangered of extinction (see Appendix A), one is endemic (Aloe woodii), and the rest are non-endemic.

Point 19: How many alien taxa are considered present in the study area? Are they invasive? Can they threaten the native flora or local ecosystems? Please discuss this topic.

Response 19:18 invasive alien taxa present in the study area, have distressing impacts on native biota. All major invasions are due to deliberate or unintentional actions of humans, birds, animals, and, to a certain extent, due to traffic across the borders. Though the majority of introductions have only minor impacts on the ecosystem.

Point 20:I suggest to add an appendix with photographs of relevant taxa and landscapes found in the study area.

Response 20:‘Actually this is a good suggestion, but unfortunately we have not photos with good quality’
